# Anti-Prion Systems Block Prion Transmission, Attenuate Prion Generation, Cure Most Prions as They Arise and Limit Prion-Induced Pathology in *Saccharomyces cerevisiae*

**DOI:** 10.3390/biology11091266

**Published:** 2022-08-26

**Authors:** Reed B. Wickner, Herman K. Edskes, Moonil Son, Songsong Wu

**Affiliations:** Laboratory of Biochemistry and Genetics, National Institute of Diabetes and Digestive and Kidney Diseases, National Institutes of Health, Bethesda, MD 0892-0830, USA

**Keywords:** anti-prion system, folded parallel in-register β-sheets, amyloid, sequestrase

## Abstract

**Simple Summary:**

Virus and bacterial infections are opposed by their hosts at many levels. Similarly, we find that infectious proteins (prions) are severely restricted by an array of host systems, acting independently to prevent infection, generation, propagation and the ill effects of yeast prions. These ‘anti-prion systems’ work in normal cells without the overproduction or deficiency of any components. DNA repair systems reverse the effects of DNA damage, with only a rare lesion propagated as a mutation. Similarly, the combined effects of several anti-prion systems cure and block the generation of all but 1 in about 5000 prions arising. We expect that application of our approach to mammalian cells will detect analogous or even homologous systems that will be useful in devising therapy for human amyloidoses, most of which are prions.

**Abstract:**

All variants of the yeast prions [PSI+] and [URE3] are detrimental to their hosts, as shown by the dramatic slowing of growth (or even lethality) of a majority, by the rare occurrence in wild isolates of even the mildest variants and by the absence of reproducible benefits of these prions. To deal with the prion problem, the host has evolved an array of anti-prion systems, acting in normal cells (without overproduction or deficiency of any component) to block prion transmission from other cells, to lower the rates of spontaneous prion generation, to cure most prions as they arise and to limit the damage caused by those variants that manage to elude these (necessarily) imperfect defenses. Here we review the properties of prion protein sequence polymorphisms Btn2, Cur1, Hsp104, Upf1,2,3, ribosome-associated chaperones, inositol polyphosphates, Sis1 and Lug1, which are responsible for these anti-prion effects. We recently showed that the combined action of ribosome-associated chaperones, nonsense-mediated decay factors and the Hsp104 disaggregase lower the frequency of [PSI+] appearance as much as 5000-fold. Moreover, while Btn2 and Cur1 are anti-prion factors against [URE3] and an unrelated artificial prion, they promote [PSI+] prion generation and propagation.

## 1. Introduction

Among the many yeast prions (infectious proteins) now known, [PSI+] and [URE3], prions of Sup35 and Ure2, have been the most extensively studied, in part because they were the first recognized as prions [1] and in part because the pioneering work of Brian Cox [2] and Francois Lacroute [3] left us with simple plate assays that connect directly with the normal functions of the prion proteins. Sup35 is a subunit of the translation termination factor [4,5], and [PSI+] cells have elevated stop codon read-through because most of the Sup35 is trapped in amyloid (reviewed in [6,7]). Ure2 is the ‘repressor’ in nitrogen catabolite repression, the shut-off of genes for utilizing poor nitrogen sources when good sources are available [8]. In [URE3] cells, Ure2 is in an amyloid state, making nitrogen catabolite repression inefficient (reviewed in [6,7]). [PIN+] is a prion of Rnq1 whose defining property is priming the generation of other prions [9,10,11].

The [PSI+], [URE3] and [PIN+] prions are in-register parallel folded beta-sheets of the corresponding protein [12,13,14]. The in-register parallel architecture is demonstrated by labeling (ideally) a single atom at a single residue in the protein, and measuring the distance between this atom and its nearest labeled neighbor in the amyloid structure. This measurement is done for many different residues in the molecule. The measurement is done by either solid-state NMR, in which case the label is a ^13^C atom replacing the usual ^12^C at a carbonyl carbon, or by electron spin resonance, in which case the label is attached as a small molecule with an unpaired electron. The NMR data show a distance close to 4.8 Å for most residues, the distance between peptide chains in a beta-sheet, showing that the structure is parallel in-register. The chemical shifts for these residues show that they are in beta-sheets. The electron spin resonance data give greater distances because the spin labels are attached and not part of the main peptide chain, but they show the same picture of a constant distance between identical residues along the beta-strand [15,16,17].

Both mammalian and yeast prions have many variants, meaning biologically and biochemically different prions based on a prion protein of the exact same protein sequence ([18,19] reviewed in [20]). A given prion variant propagates relatively stably, although mutation and segregation of mutants can occur even under non-selective conditions [21] and are common under selection (e.g., [22,23]).

The in-register parallel folded beta-sheet architecture explains why in both mammalian and yeast prions there can be prion variants [7,24] (Figure 1). That different yeast prion variants have distinct conformations of the molecule in the filament is known based on differences in D–H exchange [25] and distinguishable regions of the sequence protected from proteinase K digestion [26]. The beta-sheets are kept in-register by favorable interactions among identical amino acid side chains; such interactions would be lost if the chains went out of register. Acquiring these favorable interactions drive the monomers adding to the end of the amyloid filaments to adopt the same conformation as the molecules already in the filament [7,24]. However, the lengthwise folding of the sheet can vary from one variant to another. This folding pattern is imposed on the new monomer joining the end of the filament. This explains the templating of conformation by prions [7,24], analogous to the templating of sequence by Watson–Crick base pairing of DNA or RNA. Just recently, Caughey’s group has proven using cryoEM that fully infectious amyloid of PrP, the basis of the mammalian prions (scrapie of sheep, bovine spongiform encephalopathy and human Creutzfeldt–Jakob disease), has the same in-register parallel folded beta-sheet architecture [27]. This architecture was first shown by Tycko’s group for two amyloid variants of the Abeta peptide, the bad actor in Alzheimer’s disease [28,29]. Prion features—actual infectivity—have now been shown in some human cases of Alzheimer’s disease as well as other human amyloidoses, such as Parkinson’s disease, type II diabetes mellitus and amyotrophic lateral sclerosis, whose amyloids share this architecture (e.g., [30,31,32,33]).

The [Het-s] prion of the filamentous *Podospora anserina* was the first prion recognized to be responsible for a normal function, namely heterokaryon incompatibility, a fungal recognition of self/non-self [35]. Interestingly, the [Het-s] amyloid has a different architecture, namely, a two turns/molecule beta-helix and there is only one variant of [Het-s] (explained in refs. [36,37]). There was then an apparent strong desire among many to declare other prions to be advantageous, although the evidence was not always convincing. Finding some growth condition under which the prion was advantageous was often not reproducible and, further, would not constitute proof that an element is beneficial. For example, partial resistance to a rarely encountered chemical does not compensate for slow growth under essentially all other conditions. How can one weigh the relative advantages or disadvantages against each other over all the myriad of imaginable conditions in the yeast ecological niche?

We devised a simple method that takes a biological integral over all niches multiplied by an (imaginary) survival/growth factor. Even detrimental infectious elements, such as coronaviruses or chronic wasting disease in deer (a PrP-based prion disease) can be widespread because the infectivity may out-run the detrimental (or lethal) effects on the host. However, a beneficial infectious element will become widespread, particularly one which, like prions, cannot be geographically limited because it arises de novo at a relatively high frequency (~1 in 10^6^ cells for [PSI+] [38] or [URE3] [1]). Mitochondria and many insect endosymbionts that provide essential vitamins and amino acids are beneficial and spread as non-Mendelian genetic elements. As a result, they are ubiquitous. Accordingly, a prion that is rare in the wild must be detrimental to its host. The results are that [PSI+] and [URE3] are found in only about 1% of wild strains [39,40], and simple model computations suggest the mildest variants of these prions must have a >1% detrimental effect on growth or survival [41].

It is argued that even if a prion is usually a disadvantage, if it is occasionally an advantage, this gives the yeast a phenotypic flexibility, allowing “bet hedging” so that it can recover from a temporary bad situation with a temporary prion. However, no reproducible advantage of [URE3] or [PSI+] has yet appeared. Moreover, most variants severely slow growth, a fact not considered (neglected) by those arguing for these prions being advantageous. This makes acquiring these prions a bad bet.

## 2. Prion Protein Polymorphisms Impede Infection

Sequences of the prion domains of Sup35 and Ure2 vary more rapidly in evolution than do the remainder of each gene [42,43,44,45,46,47,48]. The ‘prion domains’ are perhaps mis-named because they have well-documented non-prion functions: the Ure2 prion domain is necessary for the stability of the protein against degradation and thus for full nitrogen regulation [49]; the Sup35 prion domain is necessary for proper mRNA turnover [50] and for cytoskeleton-associated translation [51]. These functions should constrain sequence variation in these domains.

The intraspecies variation/polymorphism of the Sup35 prion domain results in barriers to transmission of [PSI+] [52], suggesting that the mutations were selected by this effect and that the polymorphism is maintained for this reason and constitutes an anti-prion device. Similarly, the M/V polymorphism at residue 129 of PrP limits the occurrence of Creutzfeldt–Jakob disease and is believed to have been selected and maintained in the population for this reason [53]. Different *Saccharomyces* species can mate with each other to form healthy hybrids, although these hybrids show very low meiotic spore viability. Sequence differences in Sup35 and Ure2 of different *Saccharomyces* species produce barriers to transmission of [PSI+] and [URE3] [54,55].

## 3. Btn2 Sequesters Prion Amyloids, Curing Most [URE3] Isolates

Overproduction of Btn2 or its paralog Cur1 efficiently cure the [URE3] prion in a process requiring Hsp42 [56,57]. Overproduction of Hsp42 also cures [URE3] [57]. Btn2 acts by collecting the Ure2 amyloid filaments at one place in the cell so that in cell division, the probability of one of the daughter cells having no filaments (and so being cured) is dramatically increased [56]. Btn2 also collects non-amyloid aggregates [58,59,60]. Like Btn2, Hsp42 collects misfolded proteins [61], presumably including prions. Neither Btn2 nor Cur1 cure most [PSI+] variants [56,62], but both do cure an artificial prion of Nrp1 with Sup35C [59]. Cur1, although related to Btn2, does not co-localize with Ure2 in curing [URE3] [56]. It is proposed that overproduced Btn2 and Cur1 cure [URE3] by binding to Sis1 and sequestering it in the nucleus [59,63], but this model is inconsistent with the data as discussed elsewhere [64,65]. The cellular levels of Btn2 and Cur1 are extremely low [66] but are dramatically amplified by proteasome defects, resulting in curing of [URE3] [66], or by inhibition of proteasome activity [59,66]. Thus, the sequestration of misfolded proteins by (at least) Btn2 is a backup for overwhelmed proteasomes that is automatically induced [66].

We found that ~90% of [URE3] isolates arising in a *btn2*Δ *cur1*Δ strain are cured on replacing either (or both) of the normal genes [57]. This means that far more prions are arising than had been apparent but that as they arise, most of them are cured by Btn2 and/or Cur1. This situation resembles genome mutations that are known to occur at high frequency, but are largely repaired by the array of DNA repair systems. These results led us to take a similar approach to examining other curing components and to search for new anti-prion systems.

## 4. Ribosome-Associated Chaperones Cure Many [Psi+] Variants and Block Formation of Others

Ssb1 and Ssb2 (nearly identical Hsp70s), Ssz1 (Hsp70 family) and Zuo1 (Hsp40) are associated with each other on ribosomes close to the exit tunnel where the nascent protein emerges and are responsible for its correct folding ([67], reviewed in [68]). Chernoff showed that *ssb1*Δ *ssb2*Δ mutants have a 10-fold increased frequency of [PSI+] generation [69]. Replacing *SSB1* did not cure any of the [PSI+] isolates formed in the mutant, indicating that the Ssb proteins acted at the generation step of [PSI+]. The *zuo1*Δ and *ssz1*Δ mutants show a similar elevation of [PSI+] generation [70,71].

We have confirmed the earlier work on ribosome-associated chaperones, but in addition to the increased frequency of [PSI+], we found that more than half of the [PSI+] variants isolated in any of the *ssb1/2*Δ, *zuo1*Δ or *ssz1*Δ mutants are lost upon replacement of the corresponding w.t. gene, indicating that these genes block propagation of most [PSI+] variants arising [72]. The mechanism of the anti-prion effect likely involves their known role in facilitating proper folding of the normal protein, but this mechanism would suggest that they might likewise affect formation of any prion, and we were unable to detect any effect on [URE3] [72]. The proximity of the ribosome-associated chaperones to Sup35, and the small—but real—effect of their mutants on translation termination efficiency, makes possible a direct effect on the mature Sup35.

## 5. Siw14 Curing of [PSI+] and Inositol Polyphosphates

The fact that replacing Btn2 or Cur1 cured most [URE3] variants arising in a *btn2*Δ *cur1*Δ strain suggested screening the knockout collection for other genes with this ability for some prion. Targeting [PSI+], and using the *ura3-14* allele (the termination sequence from *ade1-14* inserted early in the *URA3* ORF) developed for use with the knockout collection [73], we found that *siw14*Δ strains often gave rise to [PSI+] variants that were cured on replacing the normal *SIW14* gene [74]. Siw14 is a pyrophosphatase that converts the 5-pyrophosphate group on certain inositol polyphosphates (e.g., IP7, Figure 2) to a monophosphate [75]. In *siw14*Δ strains, IP7 is elevated about 6.5-fold [75], and our results suggest that some [PSI+] variants can propagate in the presence of the higher level of IP7 but not with the lower level. We also examined the ability of an ordinary [PSI+] variant for ability to propagate in an array of mutants in the inositol polyphosphate pathway and found that 5PP-IP4, 5PP-IP4 or IP6 were each capable of supporting the prion’s propagation [74] (Figure 2). We found that none of the 24 random [PSI+] isolates tested could propagate in an *arg82*Δ mutant lacking all of the most highly phosphorylated inositol derivatives.

The soluble inositol poly/pyro-phosphates are involved in many different cellular systems (reviewed by [76,77]). Wu et al. showed that Sse1, Hsp26, Ssb1/2 and Hsp104, among many other proteins, bound to affinity columns prepared with 5PP-IP5 or IP6 [78]. Efforts to identify one of these as the explanation for the inositol polyphosphate requirement for [PSI+] have not yet borne fruit. Siw14 has a prominent role in controlling the environmental stress response (ESR) [79], but genetic evidence suggests this is not the mechanism affecting prions [74].

## 6. Normal Levels of Hsp104 Cure Many [PSI+] Variants and Decrease Formation of Others

Hsp104, cooperating with Hsp70s and Hsp40s, is critical for propagation of all of the amyloid-based yeast prions due to its ability to cleave filaments, thereby creating new growing filament ends [80,81,82,83,84,85,86]. However, overproduction of Hsp104 cures [PSI+] (and [URE3] less efficiently) [56,80,87] by a mechanism that is still controversial [88,89,90,91].

Hung and Masison isolated mutants (including *hsp104^T160M^*) of Hsp104 whose prion propagation ability and heat-tolerance were unimpaired but which had lost the ability to cure a strong [PSI+] on overproduction [92]. In an *hsp104^T160M^* strain, [PSI+] arose at ~13-fold the normal frequency, and many such [PSI+] isolates were stable in the original *hsp104^T160M^* host but unstable in a wild-type [93]. The frequency of [PSI+] variants stable in wild-type cells was also increased in *hsp104^T160M^* strains so that Hsp104 has a role in both prion generation and propagation.

There were also occasional [PSI+] isolates arising in the *hsp104^T160M^* host which were unstable in that host but became more stable on transfer to the wild-type host [93,94]. Huang et al. make the case that there are simply different variants favored by different chaperone environments. However, the dramatic increase in [PSI+] appearance frequency in *hsp104^T160M^* strains (~13-fold) shows that the net effect of the wild-type Hps104 is to impair the appearance of [PSI+], indicating that this is an anti-prion system. The fact that the frequency of variants equally stable in the *hsp104^T160M^* mutant and in the wild type is elevated [93] shows an anti-prion effect that is not only on the variants unable to propagate in the wild. Finally, the fact that there is as yet no reproducible benefit of having any [PSI+] variant makes the appellation “anti-prion” appropriate.

## 7. Upf Proteins Cure Most [PSI+] Variants by Association with Sup35

The screen for anti-[PSI+] factors that detected *SIW14* (above) also turned up *UPF1* and *UPF3* [95], encoding two of the three Upf proteins involved in nonsense-mediated decay [96]. Mutants in these genes, or *upf2*Δ, produce [PSI+] cells at ~15-fold the wild type, and restoration of the wild-type gene expressed at the normal level cures most of the variants produced in the mutant host [95]. Thus, as seen with other anti-prion systems (above), a large portion of the increased incidence of prions in anti-prion system mutants can be accounted for by the ability of variants normally cured by the given system to propagate if that system is inactive.

The Upf proteins normally directly associate with Sup35 on the ribosome, and co-localize with Sup35 amyloid fibers in [PSI+] cells [95,97], suggesting that this direct interaction might be responsible for the anti-prion effect. In fact, Upf1 interferes with the formation of amyloid by Sup35 in vitro, supporting this notion [95]. As expected, there is no effect on formation of amyloid by Ure2. Detailed mutational studies of the Upf1 helicase have ascertained functional domains and sites in the protein affecting its various activities [98,99,100]. Using this information, it was shown that it is the association with Sup35, and not the effect on nonsense mediated mRNA decay, that determines the Upf anti-prion effect [95].

The anti-prion effect of the Upf proteins is proposed to be either competition with amyloid filaments for binding to Sup35 monomers or blocking the growing ends of amyloid filaments, preventing the addition of more monomers [95]. It is likely that, in general, a protein normally interacting with a prion protein has some degree of anti-prion activity.

Upf1, along with the polyA-binding protein Pub1, also prevents the lethality of strong variants of [PSI+] [101]. Pub1 somewhat inhibited Sup35 amyloid formation, and Upf1 limited recruitment of Sup45, the other translation termination subunit, into the filaments [101]. This may be a distinct activity from the ‘anti-prion’ action above as deficiency of Upf2 or Upf3 did not produce the same effect.

## 8. Ribosome-Associated Chaperones, Nonsense-Mediated Decay Factors and Hsp104 Together Repress [PSI+] Prion Emergence ~5000-fold

Mutation of a single anti-prion system results in a 2× to 15× elevation in the frequency of prions detected, either spontaneously or following induction by overproducing the prion protein [56,57,69,72,74,93,95] (Table 1). These elevations are due to both elevated frequency of prion generation and the existence of variants that would have been cured had the anti-prion system(s) been operative. Note that in the single mutants, the other anti-prion systems are intact so that these other systems are not adequate to cure the variants detected.

We constructed strains with all combinations of *ssz1*Δ, *upf1*Δ, *hsp104^T160M^, btn2*Δ and *cur1*Δ and examined the frequency with which [PSI+] arose and the susceptibility of those variants to restoration of anti-prion systems by mating, cytoduction or introduction of plasmids with the normal genes. We found that triple mutants of genotype *ssz1*Δ *upf1*Δ *hsp104^T160M^* showed an increase of up to 5000-fold in the frequency with which [PSI+] arose [65]. We expected that the frequency elevations (about 10- to 15-fold for each of these systems) would be roughly additive since the variants sensitive to one system were evidently not sensitive to the others. That the frequency observed was at least 10-fold higher than that expected suggested that there were new variants sensitive to some combination(s) of anti-prion systems. This proved to be the case, as most of the variants isolated were destabilized by restoring any one of the systems. In addition, the last experiment proved that the Ssz1, Upf1 and Hsp104 systems worked independently of each other since restoring one system, in the absence of the other two, cured these prions.

## 9. Anti-prion Systems Turn a Tsunami of Prions into a Slow Drip

These experiments significantly change our view of the prion phenomenon. Rather than prion generation being a rare one-in-a-million event, it is clearly at least a five-in-a-thousand event (and maybe more frequent than that), with the cell eliminating nearly all of the prions arising (Figure 3). This is reminiscent of the many DNA repair systems, where nearly all of the DNA insults are repaired, with only a few making their appearance as mutations.

This new picture has several important implications. In each of our studies of anti-prion components we discovered new classes of prion variants that cannot survive long in wild-type cells because of their high sensitivity to the anti-prion component(s). There is thus an un-seen complexity of nascent prions, at least part of which is revealed in these mutants. We discussed above the infection-blocking function of prion protein amino acid sequence polymorphisms and that prions successfully passing a strong barrier are often new variants insensitive to the blockage. These altered prion variants were attributed to mis-templating because prion generation was assumed to be very rare [21] but may be due to novel variants arising de novo. When amyloid is made in vitro and its biochemical properties are correlated with biological properties after transfection, the inference may be in doubt because it is likely that only a tiny proportion of the amyloid filaments escape all the anti-prion systems to successfully infect. The number of prion variants of a given prion protein must be hard to count because variants separated by one criteria are often difficult to categorize by other criteria.

## 10. Why do Btn2 and Cur1 Affect [PSI+] The Opposite of Their Effects on [URE3]?

Btn2 and Cur1, working with Hsp42, cure most variants of [URE3] arising in a *btn2 cur1* mutant. In our study of strains defective for multiple anti-prion systems, we found that [PSI+] appearance, either spontaneous or following transient overexpression of the Sup35 prion domain, was dramatically reduced in *btn2**Δ* or *cur1**Δ* or *btn2**Δ cur1**Δ* strains [65]. Btn2 not only collects prion amyloids at one place in the cell [56] but also collects non-prion aggregates and denatured proteins [58,59,60]. Infectious amyloid of Sup35 is melted to monomers at temperatures in the 50 to 70 C range [106] while Ure2 amyloid is still unmelted at temperatures above 100 C and so is very stable [107]. Additionally, [PSI+] prions generally have seed numbers (infectious particles/cell [108]) higher than those of [URE3] prions (e.g., [56]), and normal levels of Btn2 only cure prions with low seed numbers [57]. Using these facts, we have proposed a tentative model to explain the different effects of Btn2 on [URE3] and on [PSI+] (Figure 4).

We imagine that Btn2 brings denatured Sup35 monomers to a single site in the cell, facilitating their assembly into amyloid prions. Btn2 brings Sup35 filaments to the same site as denatured monomers, helping filament elongation. However, the prions formed by Sup35 with Btn2′s help generally have seed numbers too high to be cured by Btn2. Btn2 may be also helping Ure2 prion amyloid form, but as Ure2 amyloid is much more stable, it has little need for Btn2′s help. Moreover, [URE3] prions usually have rather low seed numbers [57], and so most are cured by normal levels of Btn2.

## 11. Sis1 Reduces [PSI+] Pathogenesis by Keeping Some SUP35 Soluble (Active)

Sis1 is an Hsp40 that is necessary for cell growth but even more necessary for propagation of [URE3], [PSI+] and [PIN+] prions [109,110]. The N-terminal J domain (common among Hsp40s) and the adjacent glycine–phenyalanine-rich region are sufficient for both of these functions, without the C-terminal substrate-binding and dimerization domains, but in such Sis1JGF strains, a normally mild variant of [PSI+] is lethal [102,103,111]. Sis1JGF [PSI+] strains have lower amounts of free Sup35, and supplying a source of the Sup35 protein lacking the prion domain relieves the toxicity of [PSI+] in these cells [103]. Thus, the function of Sis1 in the Hsp104-Hsp70-Sis1 machinery that solubilizes and renatures Sup35 from the amyloid prevents the lethal effect of this normally mild [PSI+] variant. However, many [PSI+] variants are toxic or lethal in spite of a normal Sis1 and are rescued by expressing Sup35C, lacking the prion domain [23].

## 12. Lug1 Prevents Lethality of Otherwise Mild [URE3] Variants

A saturation transposon mutagenesis screen comparing [URE3] and [ure-o] hosts found that *YLR352W* could be mutated in the absence of [URE3] but not in its presence [104]. The gene, named *LUG1* for Lets [URE3] Grow, had been previously identified as a substrate-specifying subunit of a Cullin-containing E3 ubiquitin ligase [105]. However, the substrates determined by Lug1 remain unknown. Like *lug1*Δ [URE3] cells, *lug1*Δ *ure2*Δ do not grow on non-fermentable carbon sources. This defect is suppressed by either overexpression of Hap4 (a positive transcription factor stimulating transcription of mitochondrial-related genes) or by mutation of *GLN1* (glutamine synthase). The fact that *lug1*Δ [URE3] and *lug1*Δ *ure2*Δ cells have a growth defect largely specific for non-fermentable carbon sources suggest that Ure2 has an unrecognized role in carbon metabolism. Growth on proline as a nitrogen source turns off nitrogen catabolite repression, such as [URE3] or *ure2*Δ, but does not lead to a growth defect of *lug1*Δ cells, showing again that Ure2 has a role outside of nitrogen catabolite repression [104].

The same screen detected several chaperones, including those known to be needed for [URE3] propagation, but also several not yet so identified. Further work will be needed to determine the basis for their appearance in this screen [104].

## 13. Conclusions

It is evident that the cell goes to great length to limit infections by prions, the de novo generation of prions and their propagation. This constitutes yet another line of evidence that [PSI+] and [URE3] are detrimental to the host. The common human amyloidoses have recently been found to be prions, with infectious spread within each patient and, occasionally, between patients. We hope that demonstrating the existence of yeast anti-prion systems and exploring the mechanisms of their action will suggest approaches to the currently untreatable human amyloidoses based on analogous or even homologous systems.

## Figures and Tables

**Figure 1 biology-11-01266-f001:**
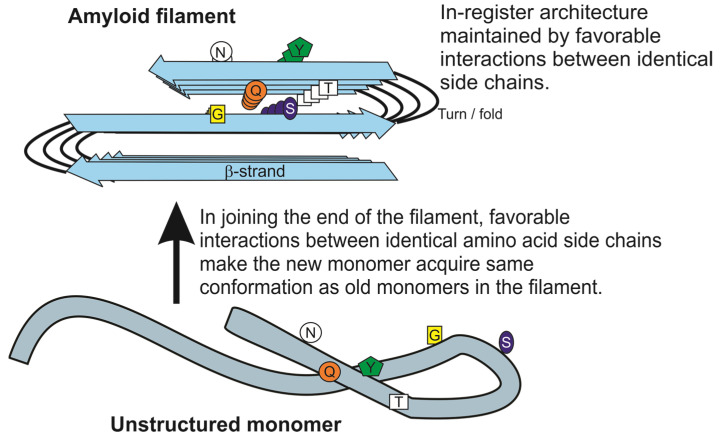
The in-register parallel folded beta-sheet architecture of yeast prion amyloids can explain their ability to template their conformation. H-bonds and hydrophobic interactions of identical amino acid side chains force adjacent molecules in the parallel beta-sheet to stay in-register and provide the fundamental force driving prions to template their conformation. A new monomer joining the end of the filament adopts a turn at the same place in the peptide chain as the molecules of that protein already in the filament in order to have those favorable side chain–side chain interactions. This is for prions what “Watson–Crick base pairing” is for DNA. The single letter amino acid code is used. Adapted from [34]. Not subject to U.S. Copyright.

**Figure 2 biology-11-01266-f002:**
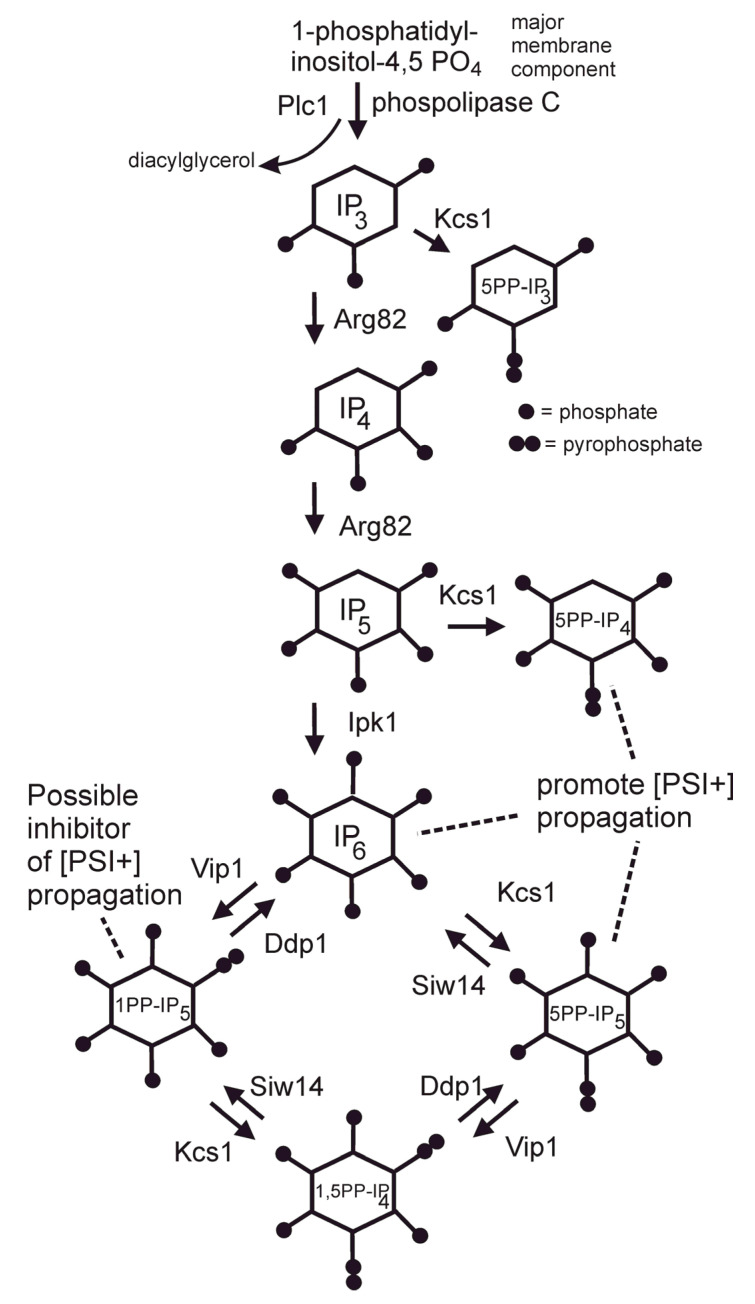
**Inositol poly/pyro phosphate metabolism.** On this diagram of the inositol polyphosphate biosynthesis pathways, the species that promote [PSI+] propagation are indicated [74]. Overproduction of 1-pyrophospo-inositolpentaphosphate (1-PP-IP5) inhibits [PSI+] propagation [74]. The mechanism by which the inositol pyro/polyphosphates affect prion propagation are unknown. Figure from [34].

**Figure 3 biology-11-01266-f003:**
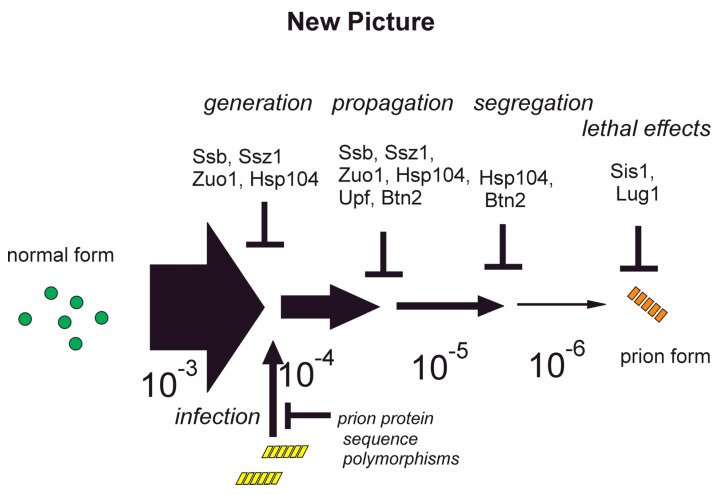
**Anti-prion systems turn a tsunami of prions into a slow drip** [65]. It was formerly thought that prion formation was extremely rare (1 in 10^6^ cells or animals) and that once formed, their propagation (and death of the animal) was assured (“The Old Picture”). The discovery of anti-prion systems and their interactions reviewed here shows that prion formation is at least 1000-fold more frequent but that only a rare nascent prion emerges. Prions are blocked at various stages.

**Figure 4 biology-11-01266-f004:**
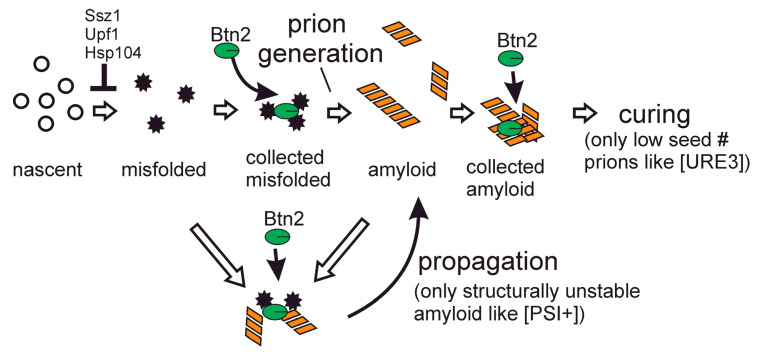
Speculative explanation for the different effects of Btn2 on [URE3] and [PSI+] [65]. Btn2 collects amyloid filaments (of Ure2 and Sup35) and non-amyloid misfolded proteins. Ure2 amyloid is far more stable and has generally lower seed number than that of Sup35. Btn2 cannot collect all the seeds of [PSI+] but by collecting misfolded Sup35 molecules may facilitate prion formation. Ure2 does not need any help to form its very stable amyloid. Thus, Btn2 sequesters the filaments of low copy number [URE3] prions but cannot collect all the Sup35 filaments. Figure from [65].

**Table 1 biology-11-01266-t001:** Anti-prion systems in yeast.

Anti-Prion Components	Target Prion(s)	Fold New Prions	Mechanism of Action	Ref.
Btn2	[URE3]	5×	Sequesters amyloid filaments and other misfolded proteins	[56,57]
Cur1	[URE3]	5×	???	[56]
Ssb1/2, Ssz1, Zuo1	[PSI+]	10–15×	Ribosome-associated chaperones that assure proper folding of nascent proteins	[69,72]
Upf1, Upf2, Upf3	[PSI+]	10–15×	Nonsense-mediated decay components that complex with Sup35 directly blocking amyloid formation	[95]
Hsp104	[PSI+], [URE3]	13×	Controversial	[80,93]
Siw14	[PSI+]	2×	Pyrophosphatase specific for 5-inositol pyrophosphates that promote [PSI+] propagation by an unknown mechanism	[74]
Prion protein polymorphism	[PSI+], [URE3],…		Intraspecies transmission barrier due to sequence difference of prion protein	[52,54]
Sis1	[PSI+]		Prevents [PSI+] filaments from depleting too much of the essential Sup35 protein	[102,103]
Lug1	[URE3]		F-box protein that prevents lethality of Ure2 deficiency	[104,105]

An anti-prion system is a set of proteins that work in normal cells (no overproduction or deletion) to block the formation of prions and cure prions as they are formed. The table includes cellular components that block prion infections and mitigate the toxicity of those prions that elude the anti-prion systems.

## Data Availability

Not applicable.

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
