# Peer review of "Anti-Prion Systems Block Prion Transmission, Attenuate Prion Generation, Cure Most Prions as They Arise and Limit Prion-Induced Pathology in Saccharomyces cerevisiae"

_biology, 2022, doi:10.3390/biology11091266_

Round 1

Reviewer 1 Report

In the manuscript «Anti-prion systems block prion transmission, attenuate prion generation, cure most prions as they arise and limit prion-induced pathology in Saccharomyces cerevisiae» authors discuss the role of different anti-prion systems in yeast prions induction and propagation. The manuscript is interesting but needs to be improved.

Traditionally, [PSI+], [PIN+] and [URE3] are written in italics with + in superscript (see e.g. Liebman & Chernoff 2012)

The designation of the protein should be uniform throughout the text. For example, Sup35 together with Sup35p are present in the text. The same concerns Ure2 - Ure2p, Sis1-Sis1p (e.g. lines 340 and 347)

Minor comments:

Lines 40-41 “ Sup35p is a subunit of the translation termination factor [4, 5]”, reference [4] is incorrect because in Frolova et al. paper the Sup45p (not Sup35p) as eRF1 was described. The paper of Zhouravleva et al., 1995 (PMID: 7664746) should be cited.

Lines 42, 45, 86 - right parenthesis omitted

Lines 112, 119 - does the underlining of words “beneficial “ and “mildest” have any meaning? The same question for the line 284 (“one”)

Line 114 – a reference(s) is needed after: “(~1 in 106 cells for [PSI+] or [URE3]).”

Lines 149-150  - repeat of “also collects aggregates” – it will be better to rephrase

Line 339 – why pathogenesis is in italic?

Line 467 - Pichia pinus – should be in italic

Reference 63 – should be corrected as “Anti-prion systems in yeast cooperate to cure or prevent the generation of nearly all [PSI+] and [URE3]  prions”

Figure 3 – it is not clear what means “New Picture”

Reviewer 2 Report

The review presented by Wickner et al. is a well-written perspective on the role of various systems in limiting the appearance and propagation of prions in yeast. I find it to be a comprehensive and illuminating perspective on the available data in the field, especially the conclusion that without the described anti-prion systems, prions would be emerging at frequencies nearing the percent range.

My only question (which the authors might or might not like to discuss in the revised version) is what thoughts do the authors have on the following: all of the described anti-prion systems have their classical non-prion-related functions, such as handling other misfolded proteins, interacting with translation machinery etc.. What are the pro- and contra- arguments on whether these systems address prions by design or by chance, simply because prions are a type of protein aggregate? As I understand, the whole evolution of individual protein sequences in general is (en masse) geared toward reducing misfolding (as described in Eugene Koonin’s book, The Logic of Chance and probably in specific papers), thus the vital role and variety of anti-misfolding systems is to be expected.

Additionally, discussion of interaction between Upfs and prions could benefit from noting that Upf is involved in reducing the potentially toxic effects of prions - 10.3390/ijms19113663.

There are also several minor textual issues, which in my opinion, were missed during the editing of the manuscript:

“The [PSI+], [URE3] and [PIN+] prions are due to the formation of in-register parallel folded beta sheets of the respective protein” – this seems a bit awkwardly phrased

“The intraspecies variation/polymorphism of the Sup35 prion domain results in barriers to transmission of [PSI+] [50], suggesting that the mutations were selected by this effect, that the polymorphism is maintained for this reason and constitutes an anti-prion device.”

In my opinion this is phrased a bit to strongly, i.e. polymorphisms might act as antiprion devices, but proof that this has been so for a specific polymorphism in yeast, as I understand, is lacking. If I am mistaken, please provide a reference. The authors do write “suggesting”, but I would say “allow for the possibility”. Also, the paragraph as a whole repeats itself (See start and end).

“We have confirmed the earlier work on ribosome-associated chaperones, but find that in addition to the increased frequency of [PSI+], we found…” - perhaps remove “find that”?

Reviewer 3 Report

This is a very well written review summarizing the research leading to discovery of the anti-prion system in yeast. This system works at the different levels to lower the rates of spontaneous prion formation, to cure prions or block their transmission from cell to cell. Table 1 informatively summarize existing information about anti-prion components, their target prions and mechanisms of action. The authors discuss their new concept supported by their recently published research paper that in cell prion generation occurs at least 1000-fold more frequent (Figure 3) then it thought before and a rare one -in-a-million event prion generation observed in normal cell is a result of action of anti-prion systems preventing escape of nascent prion from their surveillance. I certainly agree with authors conclusion that “understanding yeast anti-prion systems may facilitate discovery of analogous/homologous human systems useful in dealing with amyloid diseases”.
